# Performance of Professional Soccer Players before and after COVID-19 Infection; Observational Study with an Emphasis on Graduated Return to Play

**DOI:** 10.3390/ijerph182111688

**Published:** 2021-11-07

**Authors:** Anamarija Jurcev Savicevic, Jasna Nincevic, Sime Versic, Sarah Cuschieri, Ante Bandalovic, Ante Turic, Boris Becir, Toni Modric, Damir Sekulic

**Affiliations:** 1Teaching Institute of Public Health of Split Dalmatian County, 21000 Split, Croatia; anamarija.jurcev.savicevic@nzjz-split.hr (A.J.S.); jasna.nincevic@nzjz-split.hr (J.N.); 2Department of Health Studies, University of Split, 21000 Split, Croatia; 3Faculty of Kinesiology, University of Split, 21000 Split, Croatia; toni.modric@kifst.hr (T.M.); dado@kifst.hr (D.S.); 4HNK Hajduk Split, 21000 Split, Croatia; ante.bandalovic@hajduk.hr (A.B.); ante.turic@hajduk.hr (A.T.); boris.becir@hajduk.hr (B.B.); 5Faculty of Medicine & Surgery, University of Malta, MSD 2080 Msida, Malta; sarah.cuschieri@um.edu.mt; 6Department of Orthopedics and Traumatology, University Hospital Split, Surgery Clinic, 21000 Split, Croatia

**Keywords:** SARS-CoV-2, prevalence, football, performance, global positions system

## Abstract

The impact of the COVID-19 pandemic in sport has been the subject of numerous studies over the past two years. However, knowledge about the direct impact of COVID-19 infection on the performance of athletes is limited, and the importance of studies on this topic is crucial during the current pandemic era. This study aimed to evaluate the changes in the match running performance (MRP) of professional soccer players that occurred as a result of COVID-19 infection after fulfilling all of the prerequisites for a safe return to play (RTP). The participants were 47 professional soccer players from a team which competed in first Croatian division (21.6 years old on average) during the 2020/21 season. The total sample was divided into two subgroups based on the results of a PCR test for COVID-19, where 31 players tested positive (infected) and 16 tested negative. We observed the PCR test results (positive vs. negative PCR), the number of days needed to return to the team, number of days needed to RTP after quarantine and isolation, and MRP (10 variables measured by a global positioning system). The number of days where the infected players were not included in the team ranged from 7 to 51 (Median: 12). Significant pre- to post-COVID differences in MRP for infected players were only found for high-intensity accelerations and high-intensity decelerations (*t*-test = 2.11 and 2.13, respectively; *p* < 0.05, moderate effect size differences), with poorer performance in the post-COVID period. Since a decrease of the MRP as a result of COVID-19 infection was only noted in two variables, we can highlight appropriateness of the applied RTP. However, further adaptations and improvements of the RTP are needed with regard to high-intensity activities.

## 1. Introduction

Since the beginning of 2020, the world has been facing the global COVID-19 crisis. Different pandemic control measures with different impacts on SARS-CoV-2 transmission and population health have been widely applied [1,2]. This pandemic has affected almost all aspects of human life, causing a chain reaction in sport as well [3]. Sporting events at all levels have been cancelled or delayed, access to training facilities has been limited, and gathering opportunities have been reduced or banned, which has greatly affected fan attendance. Although most cases are asymptomatic and/or present with mild symptoms, it is well known that COVID-19 is associated with severe acute and chronic health consequences [4]. In addition, the involvement of multiple organ systems (cardiovascular, central and peripheral nervous systems, liver, skeletal muscle, and kidney) has clearly marked COVID-19 as more than a run-of-the-mill respiratory infection. 

Professional athletes are a specific subset of the population that has excellent psychophysical abilities as a prerequisite for engaging in competitive sport [5,6]. The prevalence of COVID-19 infection among this population is generally reported to be similar to that of other groups, albeit with milder clinical manifestations. Even in athletes, however, cardiac complications may occur that need to be closely monitored [7,8]. Interestingly, early pandemic studies noted an alarmingly high incidence of COVID-19 myocarditis in the general population, including among athletes (0.4–15%) [9]. Some methodological issues may explain such a wide range, such as the clinical definition of myocarditis and the screening protocols that are applied, as well as the temporal proximity of the acute phase of infection. Recently published studies among adult athletes have reported different results. Two newer large-scale studies involving 789 North American League athletes and 3018 of their colleagues who had all contracted and recovered from COVID-19 reported a low incidence (0.6% and 0.7%) of myocarditis/pericarditis. In addition, most of them suffered from a minor infection [10,11]. Moreover, both persistent and residual symptoms were observed weeks to months after the initial COVID-19 infection in athletes, such as cough, fatigue, and tachycardia, which has also observed in the general population [12]. 

Many questions remain about how the virus affects the bodies of athletes, and emerging data regarding this issue can help us to paint a clearer picture of how to deal with the infection as efficiently as possible and to return the athlete to his or her regular training regimen with minimal risks to the athlete’s health. Previous studies have mainly focused on the performance of athletes after a quarantine period (lockdown) [13,14]. A study of the physical performance in the top German and Polish national soccer (football) leagues showed that after the quarantine period, there was a significant decline in both high-intensity running and distance covered per game in Polish league matches, while in Germany, no significant differences were found [14]. A similar study was conducted on a sample of matches in the highest soccer league in Spain, and significantly higher running variables values were observed in the pre-COVID period, while on the other hand, increased values of accelerations/min and decelerations/min were noted in the post-COVID period [13]. The authors attributed these findings to congested match schedules.

Few studies have evaluated the impact of COVID-19 infection on the cardiorespiratory systems of athletes [15,16]. One such study was conducted on female volleyball players and found that pulmonary function after COVID infection was above 80% of the predicted values for each of the participants, although electrocardiography did not find signs of ischemia, arrythmias, or conduction or repolarization abnormalities [16]. The authors concluded that the studied athletes experienced the typical consequences of missing training for a period of time. In a study on 26 Olympic athletes who overcame COVID-19 with mild or moderate symptoms, magnetic resonance imaging (MR) did not find any cases of acute myocarditis, but five subjects (19%) had some kind of cardiac abnormality, with four athletes having borderline signs of isolated myocardial oedema and one with non-ischemic late gadolinium enhancement (LGE) with pleural and pericardial effusion [15].

The key questions for professional athletes are as follows: (i) When is the right time to safely return to high-intensity training and post-illness competition? (ii) What procedure should be followed? Numerous studies have attempted to assess post-COVID myocarditis and develop return-to-play (RTP) protocols. In general, RTP decision making must carefully balance the period of recommended training limitations with the risk of the adverse outcomes of COVID-related consequences, including the risk of post-rest injuries. Medical societies and sports organizations have recommended many protocols without a general consensus, leading to conflicting information for the medical professionals who are advising these athletes [8].

Recent RTP recommendations by medical societies differ in terms of the isolation or convalescence periods (7 to 14 days) in asymptomatic athletes, and the need for extensive cardiovascular risk stratification, including rest/exercise electrocardiogram (ECG), transthoracic echocardiogram (TTE), cardiac magnetic resonance imaging (CMR), and troponin level monitoring—including history of disease and physical examination according to the clinical course. A common feature of all of the protocols is gradual RTP with monitoring [17,18,19]. The clinical course, especially in the presence of fatigue, appears to affect the duration of recovery. Exercise should not be continued if the player is symptomatic, i.e., experiencing persistent fever, dyspnea at rest, cough, chest pain, or palpitations. 

Despite the importance of information about the influence of COVID-19 infection on an athlete’s physical capacities, very few studies have looked at the impact of COVID-19 infection on the working capacities of athletes [15,16]. Additionally, to the best of our knowledge, no study has thus far examined the influence of COVID-19 infection on the performance of athletes in real sport settings. This study aimed to evaluate the changes in the match running performance (MRP) of professional soccer players that occurred as a result of COVID-19 infection after fulfilling all of the prerequisites for a safe return to play (RTP). To this end, we compared the post-COVID MRP of players who had contracted COVID-19 to (i) their own pre-COVID MRP and (ii) the performance of their teammates who had not contracted COVID-19.

## 2. Materials and Methods

### 2.1. Design and Participants

This observational study was performed during the 2020/2021 soccer season in Croatia and started on 13 August 2020 and lasted until 22 May 2021. As per the regulations of Croatian national soccer (football) federation, there was no obligation to test during the season, but in the case of symptoms, a player was isolated and tested. 

The participants in this study were 47 professional soccer players from first division club (21.6 years old on average) who compete at the highest national competition level in Croatia. The inclusion criteria were active participation in soccer for at least 10 years, professional status on the team for the season (2020/2021). The exclusion criteria were musculoskeletal injury (as defined by medical staff of the team), sickness (other than COVID-19) during the observation period, and the goalkeeping playing position.

The total sample was divided into two subgroups based on the results of a PCR test for COVID-19, with 31 positive (infected—INF) and 16 negative players (noninfected—NONINF). The inclusion criteria for the NONINF group were that the player was not infected with COVID-19, did not experience any other illnesses or injuries that prevented him from participating in training or competition for more than 10 days (20 days cumulatively over the half-season), and that he played for 60 min in at least two games during the first half of the season. The inclusion criteria for the INF were that the player was diagnosed with COVID-19, that he participated in at least one game for a minimum of 60 min during the period one month before being diagnosed with COVID-19, and that he participated in at least 60 min of one game played at least one month after returning to play after COVID-19 isolation.

Players who tested positive for COVID-19 had 10 or 14 days of quarantine (regulations changed over time) and then underwent RTP protocols before starting full team training again. In detail, after 10 or 14 days of isolation (national regulative changed during time) and after a finalized standard medical evaluation [20], before becoming involved in team training sessions, players were involved in RTP, which lasted 1–4 weeks (depending on post-COVID-19 symptoms), as per recent suggestions [21,22]. RTP training sessions were based on an individual approach, meaning that players worked individually with fitness and assistant coaches according to their current fitness status. Respecting the medical rules of physical distancing, this was conducted intentionally to restore the players’ physical fitness. The focus was placed primarily on endurance and strength/power-capacities, which undoubtedly decreased due to the long period without training or being confined to home-based training [23,24]. For the asymptomatic players, home-based training was organized and monitored through online video calls and included light bodyweight strength and cardio exercises. Therefore, RTP training sessions typically included running and technical drills that aimed to improve aerobic and anaerobic capacities and technical skills. 

Specifically, on the basis of the guidelines of Elliot et al., our players initiated light aerobic work, which was gradually increased to high-intensity anaerobic work [21]. As a consequence, when the players returned to official matches, their aerobic and anaerobic endurance were most likely at their pre-sickness levels.

### 2.2. Outcomes

There are two sets of variables that are used in this study, including (i) COVID-19 variables and (ii) match running performance (MRP) parameters. The COVID-19 variables consisted of the result of the player’s PCR test (positive or negative), the number of days needed to return to the team, and the number of days needed to return to play after the quarantine and isolation period. 

MRP was measured with GPS devices (Vector S7, Catapult, Catapult Sports Ltd., Melbourne, Australia). These devices have a sampling frequency of 10 Hz, and their validity and reliability have previously been confirmed multiple times [25,26]. MRP variables included the total distance covered (m); distance covered at different speeds—low-intensity running (<14.3 km/h), running (14.4–19.7 km/h), high-intensity running (>19.8 km/h), high-speed running (19.8–25.1 km/h), and sprinting (>25.2 km/h); total accelerations and decelerations (>±0.5 m/s^2^); and accelerations and decelerations performed at high intensity (>±3 m/s^2^).

The MRP of the NONINF group were observed over the first half of the season. The MRP of the INF group were observed according to the period (pre- and post-infection). For the pre-infection period, we observed all of the games played one month (30 days) before infection. For the post-infection period, we observed all of the games played after isolation due to illness (Figure 1) 

The standard of 60 min per game (2/3 of the match duration) was based on previous suggestions [27]. It must be noted that epidemiological data were observed and reported for all players who were infected with COVID-19 (31 players in total), but MRP were only observed for those INF players who met the previously explained inclusion/exclusion criteria (13 players).

### 2.3. Statistical Analysis

The Kolmogorov–Smirnov test was used to the evaluate parametric/nonparametric nature of the variables. When the variables were normally distributed (MRP variables), means and standard deviations were presented. For categorical variables (prevalence of COVID-19 positive findings) data are presented as frequencies (counts) and percentages. For ordinal variables and the descriptive statistics, medians with interquartile range (IRQ) were included as a measure of variability. 

Differences between groups (INF and NONINF) in MRP were evaluated by a *t*-test for independent samples. Differences within group (for INF) in MRP were evaluated by a *t*-test for dependent samples. Additionally, within- and between-group differences were further analyzed using the magnitude-based Cohen’s effect size (ES) statistic with modified qualitative descriptors (ES ranges: <0.02 = trivial; 0.2–0.6 = small; >0.6–1.2 = moderate; >1.2–2.0 = large; and >2.0 very large differences) [28].

The type I error rate of 5% (*p* < 0.05) was set a priori and was considered to be statistically significant. StatSoft Statistica ver. 13.0 (Tulsa, OK, USA) was used for all of analyses.

## 3. Results

During the 2020/2021 soccer season, 57.4% (31 of 47) of the players from the observed team tested positive on COVID-19 infection. The dynamics of the positive findings are presented in Figure 1

Of the 31 infected players, 23 (74%) were symptomatic, while 9 of them (26%) were asymptomatic (tested because of contact with an infected person). Three players (10% of the infected) had poor results during their medical check, while two (6.5% of the infected) extended the adaptation period due to their poor fitness status despite satisfactory medical parameters (in both cases, the players themselves self-reported a lack of fitness at the beginning of the RTP period. 

The number of days that the infected players were not included in the team ranged from 7 to 51 (median of 12, IRQ = 13), while the number of days where the infected players did not play any games ranged from 7 to 97 (median of 13, IRQ = 33). The period of only seven days occurred the most often (24.1% of cases) and was largely caused by the match schedule and the importance of individual players.

Table 1 presents descriptive statistics, *t*-test results, and ES differences between groups (INF vs. NONINF) for their MRP before the INF players were positively diagnosed for COVID-19. In brief, the players who later suffered COVID-19 infections (INF group) achieved a higher number of high-intensity accelerations than the NONINF groups (*t*-test = 2.39, *p* < 0.01).

For most of the ES differences in the variables between groups were trivial to small, while moderate ES was evidenced for high-intensity accelerations (Figure 2). 

Descriptive statistics and *t*-test differences for the INF group (comparison of pre-COVID and post-COVID performances) are presented in Table 2. Significant differences were found for high-intensity accelerations (*t*-test = 2.11, *p* < 0.05) and high-intensity decelerations (*t*-test = 2.13, *p* < 0.04). Evidently, both performances were lower after return to play than they were before the players were isolated due to COVID-19 infection.

Figure 3 presents the ES differences for the same comparison (pre-COVID vs. post-COVID for infected players). Moderate ES differences were evidenced for high-intensity accelerations and decelerations, while trivial to small ES differences were found for the remaining MRP.

Table 3 presents descriptive statistics and *t*-test differences for the comparison of MRP between the NONINF and INF players after return to play. 

No significant difference in MRP between NONINF, and post-COVID status of INF players were evidenced (Table 3), with trivial-to-small ES differences for all variables (Figure 4).

## 4. Discussion

The results evidenced several important findings, which will be discussed in the following text. First, in the observed professional soccer team, a high prevalence of COVID-19 infection was found. Second, we can emphasize the large variations in return-to-play for those players who suffered from COVID-19 infection over the observed season. Third, according to observed MRP, the return-to-play protocols resulted in the proper recovery of the infected players.

### 4.1. COVID Prevalence

This study observed the competitive 2020–2021 soccer season, in which 57.4% of athletes on a Croatian first division soccer team was diagnosed with symptomatic or asymptomatic SARS-CoV-2 infection as confirmed by an RT PCR test. The prevalence of COVID-19 was higher among these athletes than it was in the Croatian population during the study period (<9%) [29]. Additionally, according to previously published studies on professional athletes and on soccer players in particular, indicated a lower burden of COVID-19 (less than 5%) [30,31,32]. 

A prospective cohort study on the players and staff from the German Bundesliga in 2020 showed similar seroprevalence from 1184 players and staff in May (1.99%) and June (2.09%) [30]. In a study on two teams from Denmark’s top two soccer leagues, the players were tested weekly for 11 consecutive weeks. A total of 748 players were tested, and the rate of positive COVID-19 tests was 0.53% [31]. Finally, a similar study conducted on players from Qatar’s professional soccer league found 85 positive tests from a total of 1337 players, staff, and officials (0.63%) [32].

Some methodological issues may explain the large difference between our findings (57% prevalence) and the findings from other studies conducted with soccer players (prevalence of less than 3%), such as the dominant circulating SARS-CoV-2 strain at the time of the study, different prevalence of COVID-19 in the general population, case definition, inclusion of staff members in other studies, and the duration of the study period. The high prevalence in our participants compared to other studies is likely due to the low compliance of players and staff with the control measures proposed in detail by epidemiological experts and team physicians, which were updated several times according to the epidemiological situation. The authors of the study were directly involved in team management and frequently witnessed the players and staff failing to follow the defined control measures, which undoubtedly resulted in a higher prevalence in the studied sample. Furthermore, during the study course, vaccination against COVID-19 was not mandatory for professional players in Croatia, and when the vaccination program began (December 2020), athletes were not among the targeted vaccination groups (although after some time, all professional players were offered a voluntary vaccination). In addition, a major outbreak of COVID-19 in the studied players appeared at the beginning of 2021, when the SARS-CoV-2 Alpha variant was the dominant strain circulating in Croatia, which is known to be more contagious than the original strain (i.e., note that previous studies regularly observed earlier periods when other variants of the virus were circulating). This high virus transmissibility, coupled with the low practice of control measures, both non-pharmaceutical and vaccination, is likely to have caused this outcome.

### 4.2. Return to Play

There are many important aspects of COVID-19 in sport, but given that the virus has emerged recently, practical knowledge is rather limited. Numerous studies have been published with the aim of giving instructions and guidelines on the treatment of athletes during and after infection [18,19,21,33,34]. As mentioned in the introduction, RTP protocols have been developed and were created based on prior knowledge of viral infections that attack the cardiorespiratory system [21,34]. However, there is a clear lack of research on concrete examples of infected athletes. The importance of this issue is emphasized by Impellizzeri et al., who noted that “providing specific recommendations in this situation can be challenging and generic advice is probably not very helpful because too many unpredictable individual and contextual factors can have an impact” [35]. On the other hand, more evidence-based recommendations should be provided, which should not be looked at as general principles because of the heterogenous contextual factor, but because sharing information based on real data on athletes can help build a body of knowledge and experience that can help in similar future situations [35]. Therefore, we analyzed the collected data on elite soccer players in order to provide some concrete examples of performance after COVID-19 infection in professional sport. 

As mentioned in the Section 3, the 7-day adaptation period occurred the most often. This was certainly not an ideal scenario because in that case, the players lacked team training (for more details please see the next part of the Section 4). The median value was 12 days, suggesting that after the RTP protocol, players had a certain number of days for team training and full adaptation. Some other objective circumstances should be considered when analyzing these data. Thus, for example, four players (13% of those infected) were positive just before the end of the autumn half-season, so after their period of isolation, there was a break, which artificially increased the period until their first game reappearance.

To the authors’ knowledge, the only study that analyzed the time to return to full training after COVID-19 infection was conducted on a sample of athletes competing at varying levels in different sports who had either been diagnosed with COVID-19 or some other respiratory infection [12]. The subgroup that was infected with COVID-19 had a median of 30 days to RTP, while the other group had significantly less, only 10 days. The reason for the evidently larger difference in days to RTP in the previous study compared to our data (30 days vs. 14 days, respectively) may be because of the heterogeneity of the sample in the other study, which examined athletes from different sports and competitive levels [12]. Moreover, from the viewpoint of COVID-19, it is important to emphasize that the current competitive regulations and calendar in Croatian soccer did not allow for a longer recovery period. Specifically, the players in our study had less available time to adapt to the full training process due to the competition calendar. The national soccer association allowed matches to be postponed if more than seven players were infected, but in the case of individual infections, matches were played on schedule, which in some ways, forced professional teams to speed up the return to full training and competition rhythm.

### 4.3. Match Running Performance in Pre-COVID and Post-COVID Period

A comparison of the players’ running performance in the matches played before and after COVID-19 infection indicated similar values of total distance covered, distances covered in different speed categories (e.g., low-intensity running, running, high-intensity running, high-speed running, and sprinting), total number of accelerations and decelerations, and high-intensity decelerations. In contrast, we found a significantly lower number of high-intensity accelerations in matches played after COVID-19 infection. Although we cannot prove with certainty that these results are due to COVID-19 infection, we believe that the observed performances are at least partially influenced by the specific return-to-play protocol (RTP), in which infected players participated after isolation (please see Section 2 for more details about RTP). 

Recent studies have reported that high-intensity distance covered (e.g., high-speed running and sprinting) in soccer matches corresponds to the intensity of the anaerobic threshold, while low and moderate match activities (e.g., total, low intensity and running distance covered) correspond to the aerobic threshold [36]. Therefore, high levels of aerobic and anaerobic endurance (reached in individually created RTPs) most likely enabled the players to recover their pre-COVID match running performance (MRP). This may explain the lack of differences between the total distance covered and distance covered in different speed categories (e.g., low-intensity running, running, high-intensity running, high-speed running and sprinting) between pre- and post-COVID matches.

It is clear that individually created RTPs enabled the players to reach their pre-COVID level for some MRPs, and from that perspective, RTP was proven to be properly designed. However, such individual approaches may be especially challenging in a multifaceted competitive team sport such as soccer, as they lack soccer-specific movement patterns [23]. This is in part because the general medical rules suggested physical distancing, whereby 2 m between players had to be maintained to prevent saliva droplets from coming into contact with other players [23]. Therefore, the players did not participate in soccer-specific group drills such as small-sided games. Since such drills are performed in small areas, players are limited in reaching higher running speeds and higher overall distance. Meanwhile, such drills are mostly characterized by higher acceleration rates, especially by high-intensity accelerations, which enable players to gain a spatiotemporal advantage over the opponent in small areas [27,37]. As a consequence of not being exposed to such soccer-specific patterns for a prolonged period of time (i.e., the period of isolation and the period of RTP), the players probably did not train enough using soccer-specific movements. 

In detail, asymptomatic players were not included in group-based soccer-specific training (GBSST) for at least three weeks (i.e., four weeks of isolation and one week of RTP), while symptomatic players did not participate in GBSST for 4–6 weeks (i.e., two weeks of isolation and 2–4 weeks of RTP). During this period, the player workload was about 20–40% that of the normal competitive period, as the recommended time for returning to full team training without a high risk of injury was estimated to be 3–5 weeks [23,38]. However, since RTPs were conducted during the competitive phase of the season, this was probably not fully respected. Therefore, rapid shifts from non-soccer-specific training sessions to soccer-specific movements in matches most likely did not allow players to reach adequate levels of high-intensity accelerations in post-COVID matches. This resulted in significantly fewer high-intensity accelerations in post-COVID matches compared to in pre-COVID matches.

The idea that training activities influence match performance has been shown in previous studies that have analyzed the association between running performance in training and matches. Briefly, Modric et al. demonstrated that the number of high-intensity accelerations during weekly training sessions reflects high-intensity accelerations in matches [39,40]. The authors reported moderate correlations between high-intensity accelerations in training and matches, indicating a reasonable association between them. Since a greater number of high-intensity accelerations directly affects real game performance (for some playing positions) [41], the training sessions in RTP should be more focused on utilizing a greater number of high-intensity accelerations.

### 4.4. Strengths and Limitations

Although there are numerous studies on the topic of COVID-19 in sport, there are not many investigations that compare the physical performance of players pre- and post-COVID-19 infection, as was the case in this study. The participants consisted of professional soccer players competing in the highest soccer levels in Croatia. Additionally, the instruments used to evaluate the players’ running performance were previously validated GPS devices that guarantee precise measurements of a player’s movements.

One of the limitations of the study is the fact that the observed players are members of a single team competing in the Croatian league. Therefore, care should be taken when generalizing the results and conclusions to the general soccer population. Additionally, the measurement of the MRP was not absolutely standardized since different players were observed during different games (due to differences in when players tested positive COVID-19). Therefore, it should be clearly emphasized that the specific situations during the season in terms of tactics, opponents, results, etc., could have affected the process of including players in the training process and their selection for particular games. Finally, as there is no accepted methodology regarding the time frame in evaluating post-infection performance, the authors took an arbitrary period of one month (30 days) for the analysis. However, there is no doubt that during this period, other factors could affect running performance, and therefore, this should be observed as an important limitation of the research. However, despite the mentioned limitations, we hope that our study will contribute to the field of research and will initiate further studies.

## 5. Conclusions

The results of this study showed that the prevalence of SARS-CoV-2 among the examined soccer players was evidently higher than it was in the general population and also compared to soccer players of the same competitive rank from other countries. Given the high intensity of the soccer season, condensed match schedule, and the fact that an infection removes an individual player from team training for a certain period, it is necessary to continue to work on protection measures for the mentioned population through both non-pharmaceutical measures and through vaccination in order to decrease infection rates.

One of the major findings of this study was that there was no significant decline in MRP in soccer players due to COVID-19 infection, as changes were only noted in high-intensity accelerations. Although player performance was not assessed immediately after quarantine and before the application of RTP itself, previous studies and practical findings have suggested the application of the protocol after viral infection and before involving players in game level intensities. Given that the players went through RTP and gradually became involved in team training and competitive rhythms after their isolation period, we can conclude that the RTPs that were used are relatively well designed. This also applies to the implementation time of the RTP protocol itself, as for this sport and at this level of competition, a period of seven days has generally been proven to be sufficient for adaptation. In more detail, it took approximately 3 weeks for asymptomatic and 4–6 weeks for symptomatic players to return on their pre-COVID MRP levels. However, since there is a decrease in high-intensity actions after a prolonged absence from sport-specific team training, it is clear that certain adjustments to the RTP protocol are needed to further minimize the decline in MRP, but this is even more important in order to ensure the safety of the players. Specifically, it is necessary to include individual sports-specific exercises that will provoke maximum rapid acceleration and thus further adapt the athlete to the team soccer training and match requirements.

However, it should be emphasized that generalizations are undesirable and that each infected athlete should be approached individually, taking this and previous knowledge on the topic as general guidelines and not as strict instructions.

## Figures and Tables

**Figure 1 ijerph-18-11688-f001:**
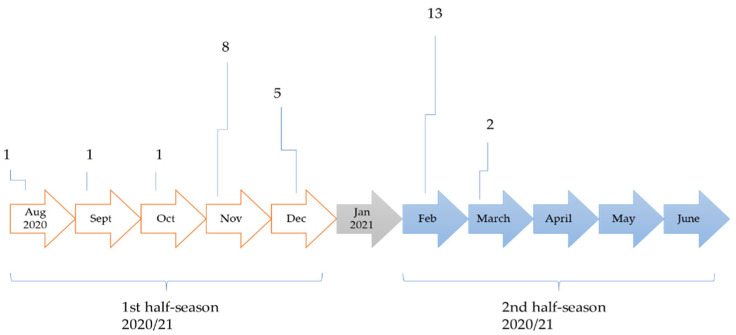
Positive COVID-19 findings during the competitive 2020/21 season in studied soccer team.

**Figure 2 ijerph-18-11688-f002:**
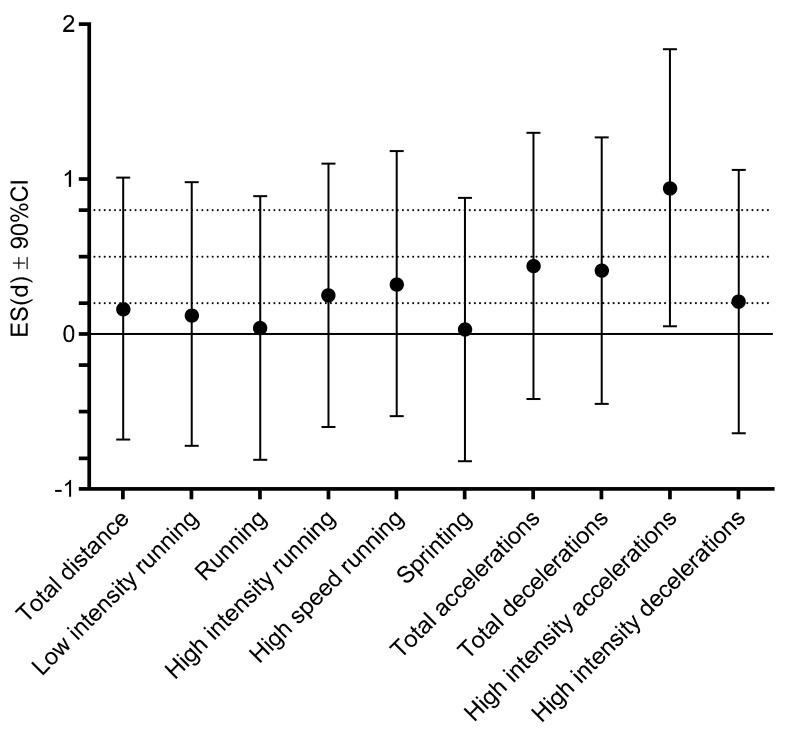
Effect size (ES) differences for match running performances between non-infected players and infected players before being diagnosed with COVID-19 (dashed lines present ES ranges: <0.02 = trivial; 0.2–0.6 = small; >0.6–1.2 = moderate; >1.2–2.0 = large; and >2.0 very large differences).

**Figure 3 ijerph-18-11688-f003:**
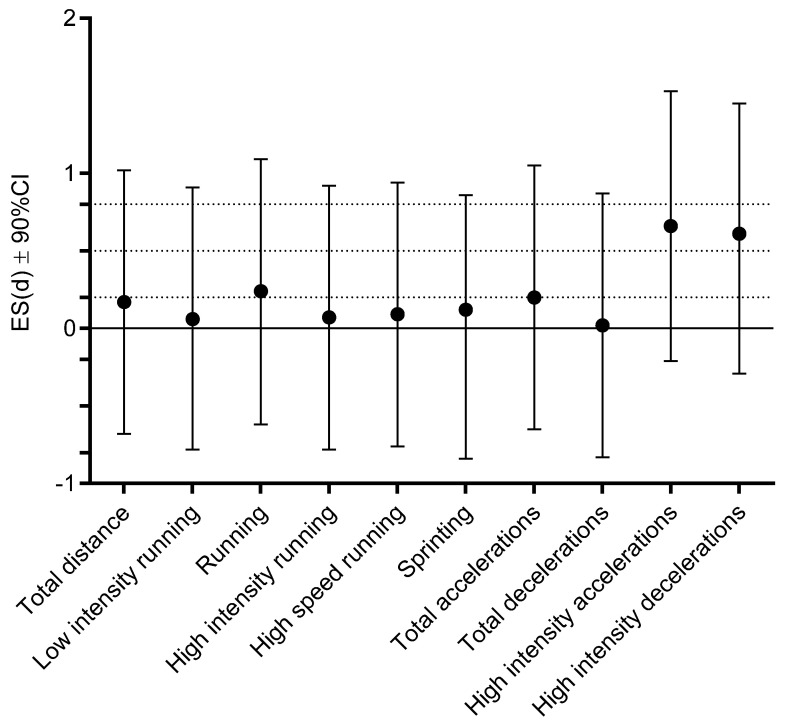
Effect size (ES) differences for match running performance within infected players before being diagnosed for COVID-19 and after return to play (dashed lines present ES ranges: <0.02 = trivial; 0.2–0.6 = small; >0.6–1.2 = moderate; >1.2–2.0 = large; and >2.0 very large differences).

**Figure 4 ijerph-18-11688-f004:**
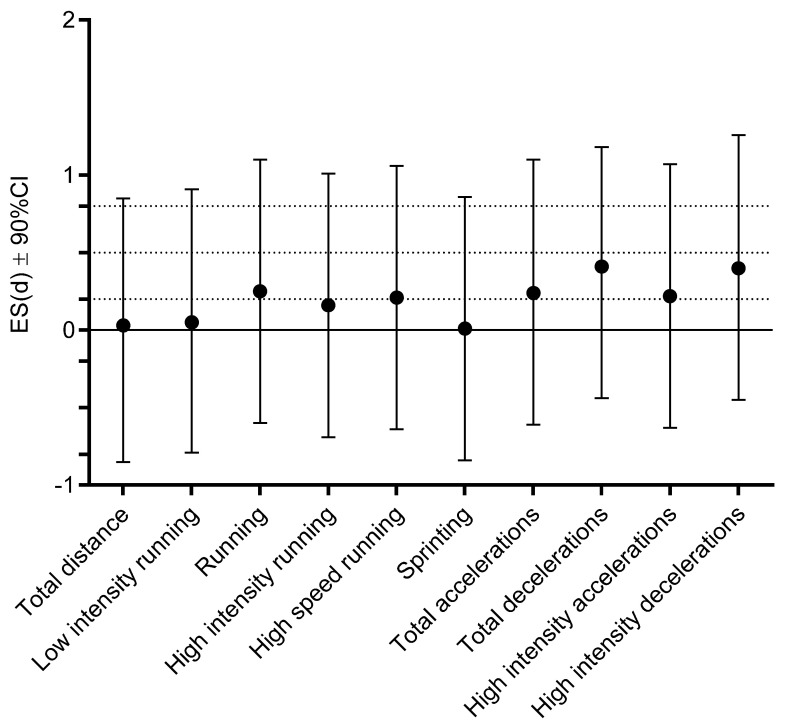
Effect size (ES) differences for match running performances between non-infected players and infected players after return to play (dashed lines present ES ranges: <0.02 = trivial; 0.2–0.6 = small; >0.6–1.2 = moderate; >1.2–2.0 = large; and >2.0 very large differences).

**Table 1 ijerph-18-11688-t001:** Descriptive statistics for match running performances and independent sample *T*-test differences between non-infected players (NONINF) and infected players before being diagnosed with COVID-19 (INF pre-COVID).

	NONINF		INF Pre-COVID		*T*-Test	
	Mean	Std.Dev.	Mean	Std.Dev.	*t*-Test	*p*
Total distance (m)	10,776.08	566.27	10,651.16	918.15	0.41	0.69
Low-intensity running (m)	8518.35	421.05	8457.23	524.94	0.32	0.75
Running (m)	1562.42	280.76	1545.55	469.29	0.11	0.92
High-intensity running (m)	697.33	197.31	648.10	193.85	0.63	0.54
High-speed running (m)	572.44	160.10	524.16	135.83	0.82	0.42
Sprinting (m)	122.07	56.76	124.22	72.55	−0.08	0.94
Total accelerations (count)	500.44	42.63	479.08	51.91	1.12	0.27
Total decelerations (count)	500.49	42.86	480.52	54.73	1.01	0.32
High-intensity accelerations (count)	18.87	8.55	28.68	11.56	−2.39	0.03
High-intensity decelerations (count)	36.54	9.35	38.10	10.34	−0.39	0.70

**Table 2 ijerph-18-11688-t002:** Descriptive statistics for match running performances and dependent sample *T*-test differences within infected players—comparison of the performances before being infected (pre-COVID) and after return to play (post-COVID).

	Pre-COVID		Post-COVID		*T*-Test	
	Mean	Std.Dev.	Mean	Std.Dev.	*t*-Test	*p*
Total distance (m)	10,651.16	918.15	10,799.96	765.13	−1.02	0.32
Low-intensity running (m)	8457.23	524.94	8490.32	519.02	−0.36	0.72
Running (m)	1545.55	469.29	1648.05	397.37	−1.12	0.28
High-intensity running (m)	648.10	193.85	662.52	232.93	−0.31	0.76
High-speed running (m)	524.16	135.83	538.72	161.38	−0.42	0.68
Sprinting (m)	124.22	72.55	123.64	87.79	0.03	0.97
Total accelerations (count)	479.08	51.91	489.63	47.98	−1.18	0.26
Total decelerations (count)	480.52	54.73	479.30	57.60	0.13	0.89
High-intensity accelerations (count)	28.68	11.56	21.22	10.83	2.11	0.04
High-intensity decelerations (count)	38.10	10.34	31.33	15.28	2.13	0.04

**Table 3 ijerph-18-11688-t003:** Descriptive statistics for match running performances and independent sample *T*-test differences between non-infected players (NONINF) and infected players after return to play after COVID-19 (INF post-COVID).

	NONINF		INF Post-COVID		*T*-Test	
	Mean	Std.Dev.	Mean	Std.Dev.	*t*-Test	*p*
Total distance (m)	107,76.08	566.27	107,99.96	765.13	−0.09	0.93
Low-intensity running (m)	8518.35	421.05	8490.32	519.02	0.15	0.88
Running (m)	1562.42	280.76	1648.05	397.37	−0.62	0.54
High-intensity running (m)	697.33	197.31	662.52	232.93	0.40	0.69
High-speed running (m)	572.44	160.10	538.72	161.38	0.52	0.61
Sprinting (m)	122.07	56.76	123.64	87.79	−0.05	0.96
Total accelerations (count)	500.44	42.63	489.63	47.98	0.59	0.56
Total decelerations (count)	500.49	42.86	479.30	57.60	1.04	0.31
High-intensity accelerations (count)	18.87	8.55	21.22	10.83	−1.00	0.28
High-intensity decelerations (count)	36.54	9.35	31.33	15.28	0.62	0.54

## Data Availability

Data will be provided to all interested parties upon reasonable request.

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
