# Peer review of "Performance of Professional Soccer Players before and after COVID-19 Infection; Observational Study with an Emphasis on Graduated Return to Play"

_ijerph, 2021, doi:10.3390/ijerph182111688_

Round 1

Reviewer 1 Report

First, I thank the editors for the opportunity to review this interesting study. In this study, the authors claim to have observed the effects of COVID 19 infection on performance variables in professional footballers and to have achieved a reduction in the potential implications through a specific protocol for a safe return to play. Despite the original work done by the authors, some critical limitations reduce its scientific value.

Main comments:
1) The authors state that "since the players after the isolation period went through RTPs and were gradually involved in team training and competitive rhythms, they can conclude that the RTPs used are relatively well designed." However, there is no evidence that RTP was necessary as the performance did not evaluate immediately before RTP.

2) The inclusion criteria for INF were that a player had been diagnosed with COVID-19, that he had participated in at least one game for a minimum of 60 minutes during the one month prior to the diagnosis of COVID-19, and who had participated in at least 60 minutes of a match played at least one month after returning to the field after COVID-19 isolation. However, in my opinion, 30 days is too long to evaluate the performance variables before and after infection (there is a similar study protocol in literature?). During this period, several variables could have influenced the performance of the players. The authors should add this remark in the limitations section.

3) The authors state that one of the study's main findings was the absence of a significant drop in MRP in soccer players due to COVID-19 infection, as changes were only seen in high-intensity accelerations.
However, even for the reduction shown in high-intensity accelerations, the Authors cannot prove that they were due with certainty to COVID-19. The authors should highlight this important consideration.

METHODS
4) should be explained the RTP protocol in the methods section.

DISCUSSION
5) Line 290-299:
Move this section to the methods and results section.
6) Line 332-348
Move this section to the methods section. 

Author Response

First, I thank the editors for the opportunity to review this interesting study. In this study, the authors claim to have observed the effects of COVID 19 infection on performance variables in professional footballers and to have achieved a reduction in the potential implications through a specific protocol for a safe return to play. Despite the original work done by the authors, some critical limitations reduce its scientific value.

Response: Thank You very much for your review and suggestions. All suggestions have been accepted and below You can find responses for specific comments.

Main comments:

1) The authors state that "since the players after the isolation period went through RTPs and were gradually involved in team training and competitive rhythms, they can conclude that the RTPs used are relatively well designed." However, there is no evidence that RTP was necessary as the performance did not evaluate immediately before RTP.

Response: Thank You for your comment. The text is amended and now reads: “Although players’ performance was not assessed immediately after quarantine and before the application of RTP itself, previous studies and practical findings have suggested the application of the protocol after viral infection and before involving players in game level intensities. Given that players after the period of isolation went through RTP and gradually became involved in team training and competitive rhythms, we can conclude that the RTPs used are relatively well designed.”

2) The inclusion criteria for INF were that a player had been diagnosed with COVID-19, that he had participated in at least one game for a minimum of 60 minutes during the one month prior to the diagnosis of COVID-19, and who had participated in at least 60 minutes of a match played at least one month after returning to the field after COVID-19 isolation. However, in my opinion, 30 days is too long to evaluate the performance variables before and after infection (there is a similar study protocol in literature?). During this period, several variables could have influenced the performance of the players. The authors should add this remark in the limitations section.

Response: Your observation is absolutely correct. We are fully aware that various factors could affected running performance but since there is no “golden standard” methodology in this kind of studies, and knowing the lack of similar researches evaluating post-infection players performances, authors unanimously concluded that period of one month (30 days) is suitable for this kind of analysis, taking account of match schedule and several other factors.  This limitation is added in Strengths and Limitations sub-section: “Finally, as there is no accepted methodology regarding the time frame in evaluating post-infection performance, authors arbitrary took period of one month (30 days) for the analysis. However, there is no doubt that  in this period other factors could affect running performance and therefore this should be observed as important limitation of the research. However, despite the mentioned limitations, we hope that our study will contribute to the field of research and initiate further studies.

3) The authors state that one of the study's main findings was the absence of a significant drop in MRP in soccer players due to COVID-19 infection, as changes were only seen in high-intensity accelerations. However, even for the reduction shown in high-intensity accelerations, the Authors cannot prove that they were due with certainty to COVID-19. The authors should highlight this important consideration.

Response: Thank You. The text is amended accordingly and now reads: “Although we cannot prove with certainty that these results are due to COVID-19 infection, we believe that observed performances are at least partially areinfluenced by the specific return to play protocol (RTP) in which infected players participated after isolation (please see Materials and Methods section for more details about RTP)” Please see end of the first paragraph of the subsection 4.3..

METHODS

4) should be explained the RTP protocol in the methods section.

Response: Thank You for your comment. Since, according to your instructions (final comment), we have moved the part of the discussion that talks about the RTP protocol to the method section, we believe that the protocol is now adequately described. (Please see Design and participants sub-section of Materials and Methods for more details).

DISCUSSION

5) Line 290-299:

Move this section to the methods and results section.

Response: Thank You, the text is amended accordingly and some parts of this section is moved to the methods and some to the results section. (Please see Materials and Methods and Results section for more details).

6) Line 332-348

Move this section to the methods section.

Response: Thank You, the text was moved according your suggestion. (Please see Design and participants sub-section of Materials and Methods for more details).

Thank you for your suggestions and comments.

Staying at your disposal

Reviewer 2 Report

I have read with interest this paper where the Authors report data regarding a crucial point on performance evaluation of professional soccer players following COVID-19 infection. For such a reason the article should be suitable for publication, nevertheless some revisions are needed.

ABSTRACT.

  • I suggest to rephase the sentence “The influence of COVID-19 infection on performance of professional athletes is rarely reported” since in the last 18 months a plethora of studies regarding this topic has been published, while I would stress the importance of research on this topic “…..is crucial point in pandemic time” or similar.

https://pubmed.ncbi.nlm.nih.gov/?term=COVID-19+infection+and+performance+of+professional+athetes

METHODS SESSION

  • Please substitute Variables with Outcomes in the subheading 2.2.

RESULT SESSION

  • Page 4 line 182-184: It is not clear why the data is reported as min and max and (median), it should be appropriate to report median and IRQ (or mean ±standard dev) and in case add min and max values.

  • Please also add in the statistical analysis paragraph how the data is presented.

DISCUSSION

Page 11 - 4.4 Strengths and Limitations: Please reconsider or remove “To the authors’ knowledge, this is the first study comparing athletes’ sport performance before and after COVID-19 infection.” From 2020 to 2021 several studies on this topic has been published.

https://pubmed.ncbi.nlm.nih.gov/?term=COVID-19+infection+and+performance+of+professional+soccer+player

Author Response

I have read with interest this paper where the Authors report data regarding a crucial point on performance evaluation of professional soccer players following COVID-19 infection. For such a reason the article should be suitable for publication, nevertheless some revisions are needed.

Response: Thank You very much for your revision. All comments are accepted and You can find responses in the text below.

ABSTRACT.

I suggest to rephrase the sentence “The influence of COVID-19 infection on performance of professional athletes is rarely reported” since in the last 18 months a plethora of studies regarding this topic has been published, while I would stress the importance of research on this topic “…..is crucial point in pandemic time” or similar.

https://pubmed.ncbi.nlm.nih.gov/?term=COVID-19+infection+and+performance+of+professional+athetes

Response: Thank You for your comment. The text is amended accordingly and now reads: “The impact of the COVID-19 pandemic in sport has been the subject of numerous studies over the past two years. However, knowledge about the direct impact of COVID-19 infection on the performance of athletes is limited, and the importance of studies on this topic is crucial in this pandemic time.” Please see beginning of the Abstract

METHODS SESSION

Please substitute Variables with Outcomes in the subheading 2.2.

Response: Thank You. Subheading is amended accordingly.

RESULT SESSION

Page 4 line 182-184: It is not clear why the data is reported as min and max and (median), it should be appropriate to report median and IRQ (or mean ±standard dev) and in case add min and max values.

Response: Thank You for your observation. We included both min and max values and median to get a clearer picture of how much the players were out of training and matches. However, as you suggested we additionally calculated the IRQ and presented it in the text. (Please see and Results section).

Please also add in the statistical analysis paragraph how the data is presented.

Response: As suggested we added text about presentation and it reads:” Kolmogorov Smirnov test was used to evaluate parametric/nonparametric nature of the variables. When variables were normally distributed (MRP variables), means and standard deviations were presented. For categorical variables (prevalence of COVID-19 positive findings) data are presented as frequencies (counts) and percent-ages. For ordinal variables descriptive statistics included medians with interquartile range (IRQ) as a measure of variability. “ Please see Statistics subsection. Thank you.

Response:

DISCUSSION

Page 11 - 4.4 Strengths and Limitations: Please reconsider or remove “To the authors’ knowledge, this is the first study comparing athletes’ sport performance before and after COVID-19 infection.” From 2020 to 2021 several studies on this topic has been published.

https://pubmed.ncbi.nlm.nih.gov/?term=COVID-19+infection+and+performance+of+professional+soccer+player

Response: Thank You. The text is rephrased and now reads: “Although there are numerous studies on COVID-19 topic in sport, there are no many investigations that compare pre- and post- COVID-19 infection player physical performance as in this study.”

Thank you for your suggestions and comments.

Staying at your disposal

Round 2

Reviewer 1 Report

The authors responded satisfactorily to the reviewer's comments; corrections have been made to the manuscript, and the quality has been implemented; therefore, the manuscript should be considered for publication.